# Screening of Toxoplasmosis in Owned and Stray Dogs of District Faisalabad, Pakistan through Latex Agglutination and Indirect ELISA

**DOI:** 10.3390/pathogens11111307

**Published:** 2022-11-07

**Authors:** Muhammad Saqib, Muhammad Sohail Sajid, Sabir Hussain, Hafiz Muhammad Rizwan, Khurram Ashfaq, Sadia Ghazanfer, Asif Ali Butt, Mahvish Maqbool, Sibtain Ahmad, Olivier Andre Sparagano

**Affiliations:** 1Department of Clinical Medicine and Surgery, University of Agriculture, Faisalabad 38040, Pakistan; 2Department of Parasitology, University of Agriculture, Faisalabad 38040, Pakistan; 3Department of Infectious Diseases and Public Health, Jockey Club College of Veterinary Medicine and Life Sciences, City University of Hong Kong, Kowloon, Hong Kong 999077, China; 4Section of Parasitology, Department of Pathobiology, KBCMA College of Veterinary and Animal Sciences, Sub Campus UVAS, Lahore 51600, Pakistan; 5Riphah International University, Faisalabad Campus, Faisalabad 38040, Pakistan; 6Institute of Animal and Dairy Sciences, Faculty of Animal Husbandry, University of Agriculture, Faisalabad 38040, Pakistan

**Keywords:** stray dogs, owned dogs, toxoplasmosis, serodiagnosis, prevalence

## Abstract

Introduction: *Toxoplasma gondii* (an intracellular protozoan) causes toxoplasmosis in warm-blooded animals, including humans and dogs. The present study was carried out to investigate the seroprevalence of canine toxoplasmosis in the owned and stray populations of dogs in Faisalabad District, Punjab, Pakistan. Materials and Methods: Commercially available Latex Agglutination Test (LAT) kits were used for the screening of samples (139 stray and 150 owned), followed by confirmation through ELISA. For the statistical analyses, chi-square was used to correlate the prevalence of toxoplasmosis with various factors. Results: The overall prevalence of toxoplasmosis, determined by the LAT, was 22.5% and, by ELISA, was 21.8%. A nonsignificant association of toxoplasmosis was determined among owned and stray dogs. Among owned dog breeds, Bulldogs showed 28.30% prevalence, and among stray dogs, the highest prevalence was determined in Bhakarwal dogs (39.29%). Young and female dogs showed a slightly higher prevalence of toxoplasmosis than adults and males, respectively. Conclusions: The present study determined by LAT and ELISA in owned dogs showed the same results, while a little variation was found in the stray dogs. It is concluded that both owned and stray dogs are infected with toxoplasmosis in Faisalabad District, and based on this, it is recommended that province-wide epidemiological studies be carried out to examine the prevalence of *Toxoplasma* and develop policies in order to control toxoplasmosis.

## 1. Introduction

*Toxoplasma* is a well-adapted and global protozoan that causes a zoonotic disease, toxoplasmosis. It is prevalent all around the world and can infect a large variety of hosts, such as mammals, birds, and reptiles [1]. *Toxoplasma gondii* is the causative agent of zoonotic toxoplasmosis, causing health issues, i.e., infertility, abortion, stillbirth, neurological, sight disorders, and death. Toxoplasmosis can be spread through the ingestion of contaminated food and water with infectious oocysts and/or that of tissue cysts (tachyzoites in the meat) [2,3]. Generally, in humans, the disease is asymptomatic, but in the case of uterine infection, it may cause mental defects, hydrocephaly, and confiscation [4,5]. *Toxoplasma* (*T*.) *gondii* is an intracellular pathogen that forms cysts in the gastrointestinal tracts of both wild and owned felids. The definitive hosts of *T. gondii* include all the felidae species [6]. 

Toxoplasmosis affects a substantial number of animals in subclinical form, with varying levels of detectable anti-*T. gondii* antibodies. The infection extent in animal populations and transmission rate to humans can be measured by measuring the antibody titers in blood and milk [6]. Different studies conducted around the globe regarding the prevalence of toxoplasmosis in dogs report that the seroprevalence of toxoplasmosis ranges from 8% to 47% [7,8,9]. Transmission sources of this parasite are raw/uncooked meat and polluted environments with oocysts of *Toxoplasma* [3,6]. 

A dog can act as a mechanical carrier for *T. gondii,* and due to its close association with other animals and humans, it is a major risk for the spread of infection. Both wild and owned dogs are sentinel creatures for a survey of ecological defilement with *Toxoplasma* oocytes due to their tendency to eat and move on feline excrement [3,10,11]. Most pet owners house their cats and dogs together. This close contact between the cats, dogs, and the owners provides favorable circumstances for the spread of toxoplasmosis. Similarly, in outdoor conditions, cats and dogs share the same environment, which increases the transmission of toxoplasmosis [12]. Most studies have mainly focused on the prevalence of toxoplasmosis in owned dogs and cats. However, few studies are available to understand the role of stray dogs in the transmission of toxoplasmosis [11]. In Pakistan, still, there has been no investigation to determine the frequency distribution of toxoplasmosis in owned and stray dogs as mechanical carriers. As dogs play an important role in the transmission and spread of toxoplasmosis in animals and humans, the present study aimed to investigate the presence of antibodies of *T. gondii* in stray and owned dogs and to determine the association of toxoplasmosis with different variables such as the age, sex, and breeds of dogs. In the present study, the latex agglutination test (LAT) and enzyme linked immune sorbent assay (ELISA) are used for the detection of infection. The present study provides valuable data about dog toxoplasmosis and helps to identify its association with different variables.

## 2. Materials and Methods

### 2.1. Study Area

The study was conducted in the Faisalabad (31°25′05.10″ N and 73°04′39.27″ E) District of Punjab, Pakistan. Faisalabad lies in the rolling flat plains of Northeast Punjab at 186 m (610 ft) above sea level. Faisalabad was reorganized as a city district composed of eight Tehsil municipal administrations (TMAs) or towns as follows: Lyallpur, Madina, Jinnah, Iqbal, Chak Jhumra, Jaranwala, Samundari, and Tandlianwala.

### 2.2. Sampling Size Estimation and Sampling 

The sample size was calculated by taking into consideration the previous *Toxoplasma* prevalence of 20% [13], with a 5% desired precision accuracy and 95% confidence limit. A convenient sampling method was used for the collection of samples. To this end, client-owned dogs presented to the Veterinary Medical Teaching Hospital, Department of Clinical Medicine and Surgery, University of Agriculture, Faisalabad, Pakistan for vaccination and routine checkup were included in the study. The animal level data (age, breed, and sex) were recorded on pretested proforma, and anamnesis (where relevant) was obtained from the owner at the time of sampling. The stray dogs captured under the Trap-Neuter-Return (TNR) program with the purpose to manage and control the dog population in and around Faisalabad District, Punjab, Pakistan were included in this study. The blood samples (~3 mL) were withdrawn directly into gel-clot activator vacutainers (Improvacuter, Guangzhou, China) through the cephalic vein. The sera were separated upon centrifugation (5000 rpm for 10 min) and stored in cryogenic vials (1.8 mL: Imec, Shanghai, China) at −30°C until testing.

### 2.3. Serological Investigations

The commercially available Latex agglutination test (LAT *T. gondii*) kit (Toxoplasma IgG, Dialab, Wiener Neudorf, Mödling district, Austria) and Innovative Diagnostics (ID) screen^®^ Toxoplasmosis indirect ELISA (Multi-species) manufactured by ID Vet (Grabels, Occitanie, France) were used to determine the presence of anti-toxoplasma antibodies in the dog sera. The LAT was carried out on diluted sera in U-bottom 96-well plates [14] and agglutination titer; 1:8 was considered the cut-off for the positive samples. The ELISA was performed following the manufacturer’s instructions. 

### 2.4. Statistical Analyses

The binomial exact technique was used to compute the prevalence. The prevalence of toxoplasmosis was calculated by dividing the number of positive samples by the total number of samples examined and multiplying by 100. The chi-square test was used to determine the statistical association between the different independent variables (age, sex, and breed). The data were analyzed using the SAS (Statistical Analysis System, version 8.0). *p*-values of less than 0.05 were considered significant statistically.

## 3. Results

The overall prevalence of canine toxoplasmosis was 22.5% and 21.8%, respectively, by LAT and ELISA. A nonsignificant association of toxoplasmosis was determined among the owned and stray dogs (Table 1). Among owned dogs, bulldogs showed the highest prevalence (28.30%), and among the stray dogs, the highest prevalence was determined in Bhakarwal dogs (39.29%). Young dogs showed a slightly higher prevalence of toxoplasmosis (LAT: 23.62%; ELISA: 22.83%) than that in adults (LAT: 21.60%; ELISA: 20.99%). Overall, female dogs (LAT: 30.53%; ELISA: 29.01%) showed a significantly higher prevalence than male dogs (LAT: 15.82%; ELISA: 15.82%).

The prevalence determined by LAT and ELISA in the owned dogs showed the same results, while a little variation was found in the stray dogs. Out of 139 stray dogs, 38 (27.34%) were positive for LAT, and 36 (25.90%) were found positive with ELISA. The frequency of toxoplasmosis (determined by LAT and ELISA) in owned and the stray dogs is given in Table 1.

Among the different towns of Faisalabad District, the highest prevalence of toxoplasmosis was detected in stray dogs (LAT and ELISA = 36.36%) belonging to Jaranwala Town. However, owned dogs of Samundari Town showed the highest prevalence of toxoplasmosis (LAT and ELISA = 25%). The prevalence of toxoplasmosis in the stray and owned dogs from different towns of Faisalabad District is shown in Figure 1. 

## 4. Discussion

The prevalence of toxoplasmosis largely varies based on the intrinsic characteristics of each region and ages of the dog populations around the globe [11]. However, no information is documented regarding the disease magnitude specifically in the stray and owned dog populations in Pakistan. The presence of insufficient local information about the disease prevalence rendered this study to estimate the magnitude of the disease. 

Outbreaks of toxoplasmosis in humans have been reported in various parts of the world. The source of infection is variable, based on eating habits and culture. *Toxoplasma gondii* is a food-borne pathogen, and about 30% of the human population is infected due to the feeding habits of undercooked meat and oocyst-contaminated food [15]. Disease spread is related to the consumption of undercooked meat and other contaminated food items [16]. Tachyzoites of *T. gondii* have been isolated from the saliva, nasal secretions, urine, and vaginal mucosa of the infected food animals [17]. The consumption of contaminated food and animal products such as milk is an imperative cause of human toxoplasmosis and is gaining concern due to an increase in the consumption of sheep and goat milk in children with a cow’s milk allergy [18]. The possible transmission of toxoplasmosis in animals and humans is given in Figure 2.

For toxoplasmosis, a laboratory diagnosis was needed to confirm the presence of infection. Most of the time, the infection exhibits no pathognomonic sign. Parasitological diagnosis is confirmed through the presence of cysts in the brain of the aborted fetus. Currently, no standardized serological test is available for the diagnosis of the parasite in animals’ meat [19]. Various serological tests, such as: ELISA, Indirect Fluorescent Antibody test (IFAT), and the Sabin–Feldman dye test (SFDT), and DNA-based tests, such as: polymerase chain reaction (PCR), LAT, and loop-mediated isothermal amplification (LAMP), can be used for the diagnosis of Toxoplasma infection [20,21,22,23]. The critical point for an acquired infection is early detection of the disease for better management of the patient. Serological diagnosis needs improvement regarding the differential of an early infection from antibodies’ nonspecific responses [24]. In the case of DNA-based tests, the results are not dependent on the immune status of the patients, and the presence of *Toxoplasma* DNA in the maternal blood is considered a recent infection [21]. *Neospora caninum* and *T. gondii* have similar morphological, immunological, and genetic characteristics. The diseases caused by these two organisms also have similar neurological and reproductive characteristics. Therefore, there is a dire need to develop a diagnostic test that can clearly differentiate the *Toxoplasma* and *Neospora* infections in animals. However, no human case of *N. caninum* infection has been confirmed [25].

A variable prevalence of toxoplasmosis in dogs has been determined in different parts of the world. For example, 62% in Brazil [26], 50% in the Czech Republic [27], 51% in Turkey [28], 12% in Spain [29], 26% in Austria [30], 47% in Iran [7], and 26% in Panama [11]. This might be due to the existence of a far-reaching population of the definitive host (cats) that favors the spread of disease or provide a favorable environment for shedding of the oocysts in the feces. However, differences in the climatic conditions and locations also play an important role to maintain and spread the infection. The humid tropical environment is one of the best conditions, which maintains oocysts of *Toxoplasma* in the soil and water [31]. The use of different diagnostic methods, size of the studied samples, and origin of the samples can affect the prevalence of toxoplasmosis [32]. The higher prevalence of toxoplasmosis in dogs might be due to their rolling on cat feces or grass contaminated with the oocysts and their coprophagic habits [33,34]. In the present study, the seroprevalence of toxoplasmosis was lower than in most of the studies [7,11,26,27,28,30]; it might be due to the use of and offer of completely cooked food to dogs and comparatively less contamination of the present study area with the parasite. The comparatively higher prevalence in other studies might be due to the use of undercooked meat containing cysts and widespread contamination of areas with oocysts of *Toxoplasma*.

Similar to our findings, Lin et al. [35], Liu et al. [36], and Zarra-Nezhad et al. [8] also determined a higher prevalence rate of *T. gondii* in female dogs in contrast to male dogs. The higher prevalence in females was largely associated with the pregnancy status and stress due to persistent hormonal changes of females during and after pregnancy and during lactation [37]. However, in Iran [38] and in Taiwan [37] a nonsignificant association of toxoplasmosis with the gender was reported.

A nonsignificant association was determined between the presence of *T. gondii* and the age of the dogs. In the current study, young animals were observed to have a slightly higher prevalence rate; however, other researchers documented a higher prevalence in older animals than that in young ones [8,39,40]. This might be attributable to the lower levels of immunity in young dogs or due to their direct exposure to the oocysts. This slight difference in different age groups might be due to variable proportions of the samples selected for this investigation. However, in adult or older dogs, the possibility of a higher prevalence of toxoplasmosis might be due to the higher frequency of the exposure to *Toxoplasma* oocysts over time [3,8]. 

Similar to our study, Hosseininejad and Hosseini [41] and Zarra-Nezhad et al. [8] also found a higher prevalence in stray dogs as compared to owned ones, which might be due to the higher frequency of exposure to risk factors such as uncooked meat; infected intermediate hosts; and contaminated water, soil, and food, with sporulated oocysts in the former [42]. The uncontrolled increase in the populations of stray dogs and cats, availability of contaminated or waste food, and lack of a garbage collection system are the possible factors that can facilitate the spread of toxoplasmosis. Thus, the spread of toxoplasmosis can be controlled by improving the environmental and socioeconomic factors of a specific geographic region. In contrast to our study, Gebremedhin et al. [43] found a nonsignificant association of toxoplasmosis with both stray and owned dogs. This might be due to the exposure of stray and owned dogs to wasted food, raw meat, or undercooked meat.

In the present study, a nonsignificant association of toxoplasmosis with different study towns was observed. This might be due to very minute or no changes in the environmental conditions (temperature and humidity) of the study towns. However, Gebremedhin et al. [43] observed a significant association in the seroprevalence of toxoplasmosis with different study towns. The moist and warm environment, along with the abundance of cats, contamination of towns with oocysts, and undercooked meat containing cysts, might change the prevalence of toxoplasmosis in different towns. Geographical locations, the density of cats, and the presence of intermediate hosts may vary the seroprevalence of toxoplasmosis in the dog populations of different towns.

Environmental changes (hot and humid), management practices (keeping cats and dogs together and free movement of dogs in the contaminated areas), and poor sanitary conditions are the factors that help in the settlement, maintenance, and transmission of toxoplasma [44]. In Pakistan, the environmental conditions (such as temperature, humidity, etc.) of Punjab Province are very favorable for *T. gondii* oocysts’ viability and infectivity [45]. In rural areas of Pakistan, stray and pet animals have free access to infection sources such as birds and rodents, which might be a contributing factor in the transmission of toxoplasmosis in the ecosystem. Most of the pet owners in Pakistan offer raw meat as a feed source, in which Toxoplasma tissue cysts can remain viable for a long time [46], hence causing the spread of the infection to healthy animals. 

Epidemiological studies are essential for the risk assessment of a particular disease in a community and/or to design a preventive management policy against that disease. Stray dogs having outside access showed more prevalence as compared to the owned ones. Both LAT and ELISA are modern and reliable techniques. Conclusively, it can be stated that *T. gondii* is a significant but neglected disease in the dog population of Pakistan. Stray dogs can play an important role in the transmission cycle and the dissemination of *Toxoplasma* to other potential hosts and reservoirs. A nationwide risk assessment of the disease using a larger sample size and inclusive of urban, peri-urban, and rural communities should be launched for a better understanding of the disease, which can help with discovering an appropriate mitigation strategy and to prevent public health hazards. Metagenomics and genotyping of different strains of *T. gondii* to map the species diversity and predict the best methodology for the control of this zoonotic disease are suggested. 

## Figures and Tables

**Figure 1 pathogens-11-01307-f001:**
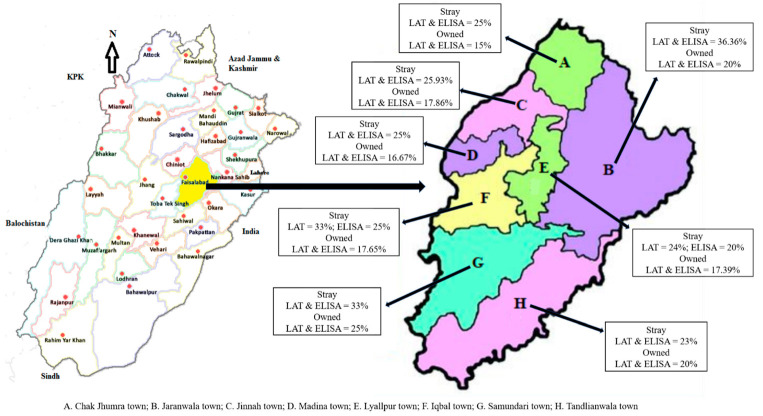
The prevalence of toxoplasmosis in stray and owned dogs of different towns of Faisalabad District. Left side is physical map of Punjab highlighting the study district of Faisalabad. Right side is the physical map of Faisalabad District highlighting different towns.

**Figure 2 pathogens-11-01307-f002:**
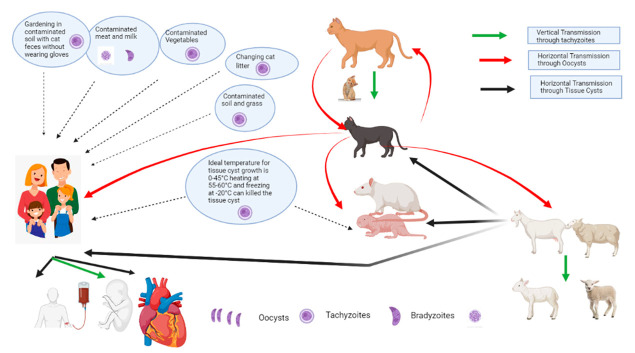
The possible transmission routes of different stages of *Toxoplasma* spp. from animals to humans.

**Table 1 pathogens-11-01307-t001:** The frequency distribution of toxoplasmosis determined by LAT and ELISA in the stray and owned dogs of Faisalabad District, Punjab, Pakistan.

Variables		Examined	Infected	Prevalence	*p*-Value	Chi-Square (χ2)	Examined	Infected	Prevalence	*p*-Value	Chi-Square (χ2)
					Toxoplasmosis in Stray Dogs						
			LAT					ELISA			
Breed	Bhakarwal dog	28	11	39.29	0.51	3.30	28	11	39.29	0.36	4.826
	Bulldog	34	9	26.47			34	9	26.47		
	Bully Kutta	39	8	20.51			39	7	17.95		
	Bhagyari dog	16	5	31.25			16	5	31.25		
	Indian Pariah dog	22	5	22.73			22	4	18.18		
Age	Young	65	15	23.08	0.29	1.12	65	14	21.54	0.27	1.210
	Adult	74	23	31.08			74	22	29.73		
Sex	Male	45	8	17.78	0.08	3.06	45	8	17.78	0.13	2.287
	Female	94	30	31.91			94	28	29.79		
Total		139	38	27.34			139	36	25.90		
					**Toxoplasmosis in owned dogs**						
Breed	Bulldog	53	15	28.30	0.20	4.69	53	15	28.30	0.20	4.69
	Greyhound	32	4	12.50			32	4	12.50		
	German Shepherd	38	5	13.16			38	5	13.16		
	Labrador	12	2	16.67			12	2	16.67		
	American Bullterrier	1	0	0.00			1	0	0.00		
	Alsatian	4	1	25.00			4	1	25.00		
	Cocker spaniel	1	0	0.00			1	0	0.00		
	Belgium shepherd	1	0	0.00			1	0	0.00		
	Rottweiler	3	0	0.00			3	0	0.00		
	Pointer	2	0	0.00			2	0	0.00		
	Mixed	3	0	0.00			3	0	0.00		
Age	Young	62	15	24.19	0.10	2.75	62	15	24.19	0.10	2.75
	Adult	88	12	13.64			88	12	13.64		
Sex	Male	113	17	15.04	0.10	2.71	113	17	15.04	0.10	2.71
	Female	37	10	27.03			37	10	27.03		
Animal keeping	Remain indoor	23	3	13.04	0.50	0.452	23	3	13.04	0.50	0.452
	Go for walk	127	24	18.90			127	24	18.90		
Total		150	27	18.00			150	27	18.00		

## Data Availability

All the data can be found in the main text.

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
