# Peer review of "Screening of Toxoplasmosis in Owned and Stray Dogs of District Faisalabad, Pakistan through Latex Agglutination and Indirect ELISA"

_pathogens, 2022, doi:10.3390/pathogens11111307_

Round 1

Reviewer 1 Report

This study investigated the prevalence of domestic and stray dogs through both latex agglutination and ELISA, as well as specificity and sensitivity for detection of Toxoplasma gondii.

Mayor comments:

1)      The authors must be determining the diagnostic performance of both tests for sensitivity and specificity, and kappa value to determine the agreement between diagnostic tests.

2)      The paper is not easy to ready, especially the discussion, where data are not compared with those found in the literature.

Minor comments:

L66: include the abbreviation of “LAT”

L93: specify the brand name of “ID screen toxoplasmosis …”

L99: first the prevalence must be determined and then the Chi-square statistical performed.

L106, replace “22.5 and 21.8 percent” to “22.5% and 21.8%”.

Reviewer 2 Report

The manuscript by Saqib and collaborators present the results of a pilot epidemiological screening of toxoplasmosis in dogs, both domestic and wild.

The idea behind is very interesting, given the epidemiological importance of dogs as reservoirs of the parasite. Indeed as a pilot study the data are sound, however the sampled population is quite low over quite a large geographical area. Nevertheless the average infection rate over the sampled area is constant, indicating an entrenching of the parasite.

In their study the Authors have used two commercial kits based on techniques that look for antibodies in the serum. Since there exists several kits, it would have been interesting to understand the reason for that specific choice. Moreover it would be useful to show a control of these kit performance on samples known to contain Neospora. The fact is that Authors make distinctions on specificity and sensitivity, without presenting data in support. Looking at what is presented in Table 1, LAT finds 2 more positive than ELISA, but in the conclusion the opposite is derived: which sentence is good?

Regarding the statistical analysis, it is not clear how it was conducted: the p-value is a binary comparison between and within samples. Here it is not at all clear where the combined number is coming from (Table 1 - column p-value and column chi-squared referred to the breed). No clear protocol is given in the methods neither.

The sentence in paragraph 123-125 is not clear, please rephrase.

Please control the typing of the names in the author list -> there are discrepancies between the manuscript and the submission

Reviewer 3 Report

The study presented has referred to the comparison of methods of analysis of prevalence of Toxoplasmosis, in the title the authors refer to public health. But I have not seen reflected the real reason for this concern, it may be because the number of inferal dogs is high in the areas under study or perhaps it is that there is little health control by the authorities regarding this group of dogs and there is a great risk of domestic dogs being infected. and why the Latex Agglutination Test method of analysis is interesting and not another.

Considering the methodology, the authors have only carried out a qualitative study and have not referred to a quantitative study to compare both methods of analysis. Perhaps the information regarding the amount of parasite detected could be expanded, and thus know if the percentage of infected dogs was really similar or not.

The title should include "pakinstan" since according to the authors there is no data prior to this article. Being able to make some comment in introduction on the environmental conditions of the area under study and thus be able to transfer your study to other regions where the conditions are similar or at least compare them as the authors in discussion have done. To say that in this comparison they did not mention the analysis methodology and I think it is necessary in this qualitative study.

And I transmit a question because the Latex Agglutination Test method of analysis is interesting and not another.

About the conclusions support the results obtained

Round 2

Reviewer 1 Report

Most of the discussion section could be in the introduction. Discuss the results obtained with the literature consulted.

Author Response

Most of the discussion section could be in the introduction. Discuss the results obtained with the literature consulted.

Agreed: The revised draft has been reviewed by experts in this field to avoid typos, English language, style, and grammatical mistakes.

From the discussion, various lines have been removed, and added introduction section line number 38-42.

The compare and contrast with other literature have been added as per your suggestions in line number 194-199, and 224-236 and relevant references have been cited accordingly.